# Correlation of Mercury Occurrence with Age, Elemental Composition, and Life History in Sea-Run Food Fish from the Canadian Arctic Archipelago’s Lower Northwest Passage

**DOI:** 10.3390/foods10112621

**Published:** 2021-10-29

**Authors:** Iris Koch, Pranab Das, Bronte E. McPhedran, John M. Casselman, Kristy L. Moniz, Peter van Coeverden de Groot, James Qitsualik, Derek Muir, Stephan Schott, Virginia K. Walker

**Affiliations:** 1Department of Biology, Queen’s University, Kingston, ON K7L 3N6, Canada; koch-i@rmc.ca (I.K.); pd50@queensu.ca (P.D.); bronte.mcphedran@queensu.ca (B.E.M.); john.casselman@queensu.ca (J.M.C.); monizk@queensu.ca (K.L.M.); degrootp@queensu.ca (P.v.C.d.G.); 2Environmental Services Group, Royal Military College of Canada, Kingston, ON K7K 4B4, Canada; 3Gjoa Haven Hunters and Trappers Association, Gjoa Haven, NU X0B 1J0, Canada; gjoa@krwb.ca; 4Environment and Climate Change Canada, Canada Center for Inland Waters, Burlington, ON L7S 1A1, Canada; derek.muir@ec.gc.ca; 5School of Public Policy and Administration, Carleton University, Ottawa, ON K1S 5B6, Canada; StephanSchott@Cunet.Carleton.Ca; 6School of Environmental Studies, Queen’s University, Kingston, ON K7L 3N6, Canada

**Keywords:** anadromous salmonids, Arctic char, lake whitefish, cisco, lake trout, isotopes, otoliths

## Abstract

As mercury emissions continue and climate-mediated permafrost thaw increases the burden of this contaminant in northern waters, Inuit from a Northwest passage community in the Canadian Arctic Archipelago pressed for an assessment of their subsistence catches. Sea-run salmonids (*n* = 537) comprising Arctic char (*Salvelinus alpinus*), lake trout (*S. namaycush*), lake whitefish (*Coregonus clupeaformis*), and cisco (*C. autumnalis, C. sardinella*) were analyzed for muscle mercury. Methylmercury is a neurotoxin and bioaccumulated with fish age, but other factors including selenium and other elements, diet and trophic level as assessed by stable isotopes of nitrogen (δ^15^N) and carbon (δ^13^C), as well as growth rate, condition, and geographic origin, also contributed depending on the species, even though all the fish shared a similar anadromous or sea-run life history. Although mean mercury concentrations for most of the species were ~0.09 µg·g^−1^ wet weight (ww), below the levels described in several jurisdictions for subsistence fisheries (0.2 µg·g^−1^ ww), 70% of lake trout were above this guideline (0.35 µg·g^−1^ ww), and 19% exceeded the 2.5-fold higher levels for commercial sale. We thus urge the development of consumption advisories for lake trout for the protection of pregnant women and young children and that additionally, periodic community-based monitoring be initiated.

## 1. Introduction

Mercury (Hg) is a persistent pollutant with neurotoxic derivatives that originates primarily from anthropogenic sources [1,2,3,4]. The photochemical oxidation of Hg^0^ to divalent mercury (Hg^+2^), as occurs during the polar sunrise in the spring, results in a reduced atmospheric residence time, and thus this reactive contaminant quickly accumulates in snow or ice and then rapidly leaches into the aquatic system during spring melt [2,5]. Hg levels in Arctic freshwaters are therefore ultimately contingent on factors such as drainage basin characteristics, with climate change predicted to increase the Hg burden in catchments and lakes [6]. The Arctic landscape is undergoing accelerated change due to thawing permafrost. With the continued atmospheric loading of Hg, as well as the impact of permafrost thaw on the global Hg cycle, it is now more important than ever to determine the accumulation of this and other elements in Arctic food fish that are seasonal freshwater residents. Do Indigenous communities that depend on these resources need to be concerned about Hg in their food fish?

The most abundant form of Hg found in fish tissue is methylmercury (MeHg), and therefore total mercury (THg) levels including organic MeHg, a neurotoxin, are most consistently reported in food fish [7,8]. The microbial-mediated conversion to MeHg increases with temperature and at low pH, further linking climate change and increasing acidification of water bodies by the uptake of CO_2_ with MeHg contamination, as well as with tissue accumulation due to rapid uptake and slow elimination [9,10,11,12,13]. MeHg biomagnifies through food webs and is often highest in predatory fish with THg accumulation, often investigated with respect to diet, size, and growth rate [14,15,16]. Fish that are slow-growing, larger, and piscivorous generally have higher concentrations of THg than fast-growing, planktivorous fish [17]. Moreover, although higher THg might be found in fattier fish, previously, we showed a negative relationship between THg and percent lipid in these salmonids [18]. THg levels also depend on fish habitat and life histories, with levels of selenium (Se) playing a role since Se/THg ratios > 1 appear to confer some protection from toxicity, likely due to the formation of insoluble Se/MeHg precipitates [19,20]. 

Common measures for normalizing levels of contaminants at size typically do not routinely consider environmentally malleable growth rates in fish, as well as variable growth between species, populations, and even individuals [21,22,23,24]. This is of concern since in the high Arctic, fish typically grow slowly and live to advanced ages, as might be expected due to generally low fishing pressure and mortality rates combined with nutrient limitations [25,26]. Under these conditions, we posit that age determined by otoliths, calcium structures of the inner ear, rather than overall length, provides a more accurate record of seasonal growth [27]. Additionally, government consumption guidelines for food fish at risk for THg-contamination, where they have been established, frequently depend on size, underscoring the importance of this investigation. Likely due to insufficient information on some salmonid species and on a community-by-community basis, there are no current advisories for Nunavut fish, including in our study region. 

Here, otolith age assessments were used to investigate THg contamination in four popular subsistence salmonids at sites on or near *Qikiqtaq*, or King William Island, in the lower Northwest passage in the east Kitikmeot region of Nunavut, Canada. *Salvelinus alpinus* (Arctic char, Inuktitut name: *iqaluk*) has the most northerly distribution of any freshwater fish [28], with the region’s sampled populations anadromous, migrating up rivers prior to sea-ice formation in the autumn and back to coastal waters to feed after the spring breakup. The region’s sampled fish in the whitefish complex, including *Coregonus clupeaformis* (lake whitefish, Inuktitut names: *kakkiviaqtuuq* and *pikugtuuq*) as well as *C. autumnalis* (Arctic cisco, Inuktitut name: *kavihilik*) and *C. sardinella* (least cisco, Inuktitut name: *anaaqtiq*), are also anadromous, overwintering in lakes but entering brackish waters on a seasonal basis, and according to Indigenous Knowledge (IK) or Inuit *Qaujimajatuqangit*, coming back to fresh water within days of the char migration peak. Although *S. namaycush* (lake trout, Inuktitut name: *ihok*) have been reported to be the least salt-water-tolerant of the salmonids [28], they also seasonally migrate to the sea in this region of the Arctic according to IK, similar to lake trout from the west Kitikmeot region [29]. All these salmonids, possibly excepting some cisco, are directly targeted for local consumption on *Qikiqtaq* and adjacent mainland sites during the autumn migration, with ice-fishing also occurring after “ice-up” and into the spring breakup. 

Residents of the *Qikiqtaq* community of *Uqsuqtuuk*, or Gjoa Haven, are regular consumers of these fish and requested that contaminants be assessed. Although THg has been intermittently assayed in select species in the Canadian north from the 1970s ([30,31,32] with some data privately available from individual researchers), no water bodies on and near *Qikiqtaq* have been systematically sampled. A single exception was a report of THg concentrations in several Arctic char sampled by the community in 2004 [6,32]. However, no values were acquired for other important subsistence fish species, leaving a knowledge gap. Thus, an investigation into THg contamination of important subsistence and potential commercial fish species in this region is not only timely but important to the resident Inuit. Our goals were to: (1) respond to the push from the Inuit fishers for THg information from other species, as well as to revisit the decades-old data from a few Arctic char, (2) use age data rather than fish size to compare THg levels in four anadromous salmonids that occupy the same migratory waters, (3) examine relationships between biological factors, stable isotope ratios, levels of selenium, and other elements with THg contamination, and (4) compare, if possible, THg levels in fish from different favored fishing sites as requested by the community.

## 2. Materials and Methods

### 2.1. Sample Collection 

The study area encompassed about 67,000 km^2^ on or adjacent to *Qikiqtaq* and south of the Adelaide Peninsula in the Kitikmeot region of Nunavut Canada. Fishing sites were selected in partnership with *Uqsuqtuuk* community fishers and elders, and were situated on migratory routes in estuaries, rivers, or in lakes with sea access. Two distinct fishing seasons of December–June (freshwater lakes and rivers under thick ice) and August–September (at open-water rivers or brackish waters prior to ice formation) were sampled over several years (2016–2019) using commercial (140 mm mesh) or multi-mesh subsistence (5 or 8 panels of 38–140 mm panels) fishing nets that were soaked for several hours. Occasionally, hand lines or spears were employed, ensuring that different populations of fish were sampled at dispersed fishing sites (Table 1, Figure 1). Licenses to fish for scientific purposes were obtained in accordance with section 52 of the general fishery regulations of the Fisheries Act, Department of Fisheries and Oceans Canada (DFO). Animal care permits were issued by the Freshwater Institute Animal Care Committee of DFO (S-18/19-1045-NU and FWI-ACC AUP-2018-63).

Fish were all assigned a unique barcode identifier [33]. Facilitated by trained community youth, fish were measured for fork length and weight, with otoliths dissected and subsequently dried. Skeletal muscles were dissected from the dorsal thorax. In a few cases, depending upon weather or logistical constraints, fish were decapitated and heads with attached muscle and otoliths were frozen whole for later processing. All muscle samples were frozen at −20 °C and shipped with freezer packs at the same temperature to the laboratory for analysis. The balance of each fish was returned to the fishers for consumption by family or other community members. 

**Figure 1 foods-10-02621-f001:**
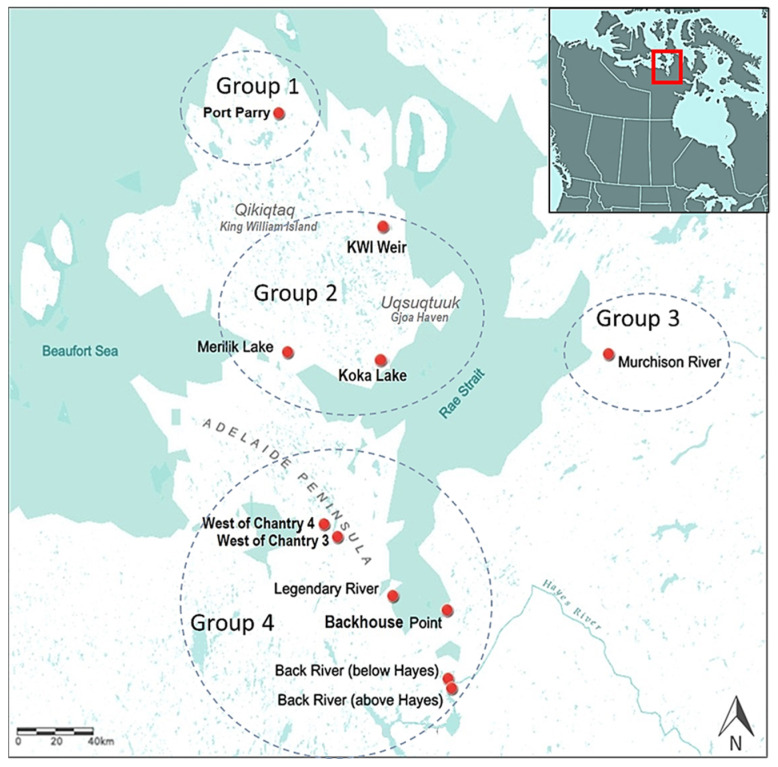
Fishing sites on or near *Qikiqtaq* or King William Island (KWI) in the Kitikmeot region of Nunavut, Canada, and showing its position on a portion of a map of North America (inset box). Fresh water sites included Port Parry, Koka Lake, Merilik Lake, Murchison River, Back River, KWI Weir, and the lakes West of Chantry. Brackish water sites included Legendary River and Backhouse Point. Note that not every fishing site is shown but all were placed into one of four groupings (shown as dashed circles). Site place names in Inuktitut are listed in Table 1. The map was created in ArcGIS online (Sources: Esri, USGS, NOAA, Garmin, USGS, NPS) [34].

### 2.2. Life History Analysis 

Age interpretation was conducted using a visual analysis of the optical zonation in transverse otolith sections. The dried left otolith was embedded in a cylinder of araldite resin and cross-sectioned through the origin [35]. A low-speed Isomet precision sectioning saw (Buehler, Opti-Tech Canada, Whitby, Ontario) with two blades separated by a spacer was used to extract a 400 µm thin section, which was mounted on a glass slide with Araldite resin, and subsequently ground and polished (Buehler EcoMet^®^; Buehler, Opti-Tech Scientific) using 400-, then 800-grit discs, leaving approximately 300–380 µm thin sections. These were microscopically viewed at 25, 40, and 100× magnification, and annuli, or growth discontinuities associated with the distal edge of translucent zones, were counted and tracked in three directions along the ventral, dorsal, and medial axes (additional details are provided in Appendix A). This method of otolith preparation and age interpretation has been validated for Arctic fish using bomb radiocarbon chronologies [36,37]. 

Fish growth was evaluated in relation to a growth standard derived from the species sampled. Each mean annual incremental growth standard was calculated using mean incremental fork length (FL) at age for each fish (mm FL∙age^−1^), with a log–log relationship developed for the entire sample. The antilog of the mean incremental value at each age provided the annual incremental growth standard curve (growth standards provided in Appendix A). Every individual was then compared to this standard and expressed as a relative difference (%), as follows: Relative difference (%) = ((FL age^−1^ − growth standard)/(growth standard) × 100(1)

Individual growth differences were then used to examine the effects of growth on other parameters. 

Otoliths not only provided year class information as indicated above, but also facilitated the identification of individual hybrid cisco individuals that are common in this region, as reported by IK and supported by our observations of fish appearance as well as preliminary genetic analysis (not shown), and as previously noted in other regions [38]. The presence of hybrids further justifies the grouping of the two cisco species for analysis. For any of a few presumptive cisco and lake whitefish mixed-heritage fish hybrids where there was disagreement as to the species, the sample was classified into one or the other fish type by otolith morphology.

### 2.3. Elemental Analysis 

Frozen muscle samples (200 mg, ~2 cm^3^) were cut into small pieces using a sterile knife and dried at 70 °C for most elemental analyses, but at 21 °C for THg and MeHg. After drying and the calculation of mean moisture content, an aliquot of each sample was ground to a powder with a methanol-cleaned mortar and pestle, and after discarding the first grindate, the second grindate (500 mg) was removed to a glass vial. THg was quantified on the solid sample using cold vapor atomic absorption spectroscopy (CVAAS) at Queen’s University Analytical Services (QUAS, Kingston, Ontario) following the United States Environmental Protection Agency (USEPA) Method 7473 [39]. Dried (21 °C) tissue samples from some lake trout were analyzed for MeHg (Flett Research Ltd., Winnipeg, Manitoba) following grinding by a ball mill, digestion with 25% potassium hydroxide-methanol and aqueous ethylation, and analyzed using purge and trap and CVAAS [40,41]. Analysis of selenium (Se) and 58 other elements, excluding Hg, was accomplished by closed-vessel aqua regia acid digestion followed by measurements using inductively coupled plasma (ICP) mass spectrometry (MS), except for boron (B), phosphorus (P), and sulfur (S), which were measured using ICP-optical emission spectroscopy (OES) at QUAS. Internal standards used to correct for plasma instability and other instrumental inconsistences included scandium, indium, and bismuth for ICP-MS and yttrium and scandium for ICP-OES. Blanks were used to monitor any cross-contamination or sources of THg, MeHg, or any other elemental contamination during the procedure, and results were less than the reporting limits for the method for each element (provided in Appendix A). Both laboratories used for the analysis are accredited by the Canadian Association for Laboratory Accreditation Inc., according to ISO/IEC 17025:2017, and the methods used for the analysis were listed on their scopes of accreditation. Species-specific mean moisture content was used to convert THg and Se dry weight concentrations to wet weight (ww) for presentation, with working standards to ensure consistency, as recommended for the methods. 

Analysis of stable isotopes of nitrogen (δ^15^N) and carbon (δ^13^C) followed previously reported guidelines [42,43,44]. Briefly, quantification was conducted on 500 mg muscle samples dried at 70 °C as described above and analyzed with a 4010 Elemental Analyzer (Costech Instruments, Italy) combined with a Delta Plus XL continuous flow isotope ratio mass spectrometer (Thermo-Finnigan, Weiler bei Bingen, Germany) at the University of Waterloo Environmental Isotope Lab (Waterloo, ON, Canada). Working standards ensured consistency among sample runs. 

### 2.4. Data Analysis 

Biological data (age, weight, condition, length, and growth), in addition to THg and Se data, were log transformed (log_10_) to achieve normal distributions (assessed with the Kolmogorov–Smirnov test of normality using a software package: IBM^®^ SPSS Statistics 25) prior to statistical analysis. Condition (K) was calculated based on Fulton’s equation (K = W·L3^−1^ [45]). Relationships between THg, age, length, weight, condition factor, and stable isotope ratios were analyzed using a Pearson correlation coefficient to determine correlated variables. Linear regressions were then used to inform the proportion of variance in the dependent variable, predictable from the independent variable (coefficient of determination, r^2^ [46]). The statistical significance of r^2^ used the F statistic by quantifying the ratio of two variances. Secondary explanatory variables (r^2^) were used to interpret interspecies differences in THg accumulation with age. Linear models for THg concentrations at a calculated standardized age were used to compare THg in fish from different sampling sites. Statistical differences in the THg means from grouped sites with unequal group sizes and heterogeneity of variances were performed using Welch’s tests with the Games–Howell post hoc test used to evaluate differences in means between grouped sites, with other statistical tests performed in IBM SPSS Statistics 25 and Statistix (version 9.0), except where noted otherwise. 

Spearman’s rank correlations, a non-parametric measure, were calculated between the elements detectable in more than 70% of samples (Hg, arsenic As, nickel Ni, rubidium Rb, strontium Sr, thallium Tl, titanium Ti, cobalt Co, chromium Cr, copper Cu, iron Fe, manganese Mn, Se, zinc Zn, calcium Ca, magnesium Mg, P, potassium K, sodium Na, and S), as well as age, length, weight, K value, δ^13^C, and δ^15^N, using XLSTAT 2020.4.1 (see Appendix A for details on detection limits of elements; Appendix A). Concentrations below the detection limits were imputed using log-normal regression on order statistics (ROS) methods in ProUCL 5.1.

To better visualize comparisons between the THg from different fish, exploratory principal components analysis (PCA) was conducted with the same variables used for the correlation analysis (observations used only those fish samples for which all but the stable isotope analysis had been conducted, *n* = 321, Spearman correlations) using XLSTAT 2020.4.1. Again, concentrations below detection limits were imputed with ROS in ProUCL 5.1. 

## 3. Results and Discussion

### 3.1. Comparison of THg Concentrations in Relation to Life History Characteristics

Mean THg concentrations for both lake whitefish and cisco at 0.09 µg∙g^−1^ ww (Table 2) were consistent with previous reports of 0.11 µg∙g^−1^ ww for *Coregonus* species across northern Canada [30]. Mean THg levels for both *Salvelinus* species were also in accordance with that large dataset, as well as more regionally with Arctic char at 0.07 vs. 0.04 µg∙g^−1^ ww, and anadromous lake trout at 0.35 vs. 0.12–0.21 µg∙g^−1^ ww, as reported here and from Nunavut’s west Kitikmeot, respectively [32,47,48]. Nets, hand lines, and spears employed by Inuit community members for sampling fish (Table 1) explain why very small, young fish were not sampled, but since the community interest was in a potential dietary exposure to Hg, the methods used were deemed appropriate. THg levels were assumed to represent MeHg and this was tested by directly assaying MeHg in randomly selected lake trout samples. These assays (*n* = 6, from different fishing sites) were consistent with previous reports showing that 95% of the THg was contributed by MeHg, which is known to be derived from dietary sources in addition to only a minor aqueous component from respiration [7,49]. 

We considered that between-species THg variation could be due to differences in growth rate and trophic level. It has been reported that slow-growing fish bioaccumulate greater concentrations of THg than do fast-growing fish (e.g., [15,17,50], but see a dissenting view with data for two species [26]). Other studies nuance this correlation and show that fish consuming pelagic prey have higher THg concentrations than those that eat benthic species [51,52]. Bioaccumulation of THg in predator fish is also positively correlated with fish weight [20]. Considering the parameters of the von Bertalanffy growth relationship, Arctic char and lake trout reach a similar ultimate length, or L∞, of 758 and 770 mm, respectively (with L∞ as the mathematical estimate of the size fish would reach if they were to live and grow indefinitely). Lake whitefish and cisco have lesser but similar ultimate lengths of 423 and 429 mm, respectively. However, fork length and weight contributed to THg variation only in lake trout and lake whitefish (Table 3, Appendix A), but not in cisco, as was reported for Alaskan-caught fish [53]. These results show that these growth parameters by themselves are insufficient to explain THg concentrations. 

Fork lengths are frequently used as a proxy for age, but as indicated, at high latitudes, fish grow slowly, and in our samples, growth leveled off after ages 5–10, underscoring the importance of using otolith year class for accuracy (Appendix A). Lake trout (*n* = 136) had the greatest age range (8–62 years), with a mean age of 25.4, just higher than lake whitefish (*n* = 100), with a mean age of 21.3 (range of 4–47 years) (Table 2). Arctic char (*n* = 197) had a younger mean age of 14.2 (5–29 years). Both species of cisco were pooled (*n* = 98), since as noted, the presence of hybrids made identification to species more challenging, and they had a mean age of 14.8 years (2–30-year range). Of the factors influencing THg, age as determined by otoliths was the only variable that was significant for all examined fish (Table 3), consistent with previous observations that age was the strongest predictor in two of these Arctic salmonids [46], but here we have expanded this correlation to all four salmonids. Age had the greatest influence on lake whitefish THg, significantly explaining 39% of within-species variation (F(1, 99) = 62.03, *p* < 0.01), with age also explaining 30% of the variation in lake trout (F(1, 135) = 57.36, *p* < 0.01). Although not as strong, age explained 14% of the Arctic char THg variation (F(1, 196) = 32.5, *p* < 0.01) and 12% of the variation in cisco (F(1, 97) = 12.72, *p* = 0.001). This association was further confirmed when the salmonids were pooled and analyzed together, showing a positive rank correlation between THg and age (Spearman test, DF = 319, *p* = 0.56), as shown in Figure 2.

The otolith age data, used to develop an incremental growth standard for each of the fish groups (Appendix A), showed no deviation from growth standards for ocean-caught and freshwater-caught fish from this region, as previously reported [54,55]. Notably, growth analysis showed that over the sample age range, Arctic char were uniformly faster growing, followed by lake trout, with cisco and lake whitefish being similar, but with the latter attaining a considerably greater age than the former (Appendix A). It has been reported that there can be some within-species variation in sub-Arctic fish due to different morph types [51,56], but different morphs were not present in our samples. Annual incremental growth-at-age data were consistent within all four salmonid groupings (Appendix A). Since lake trout are of the greatest concern for levels of this contaminant, it is important to note that there have been no reported differences in THg with respect to life history, either resident or migratory [47]. The high growth rate of Arctic char compared to the other salmonids likely contributed to their overall low THg levels, as supported by the negative relationship between Arctic char growth and THg, not seen in lake trout (Table 3). Indeed, concentrations of the contaminant were significantly different between Arctic char and lake trout of equal size (two-sample T-test, F(38.9), *p* < 0.000). Tying low THg levels with rapid growth is not adequate by itself, however, when one considers that lake trout had the second highest growth rate but accumulated the highest THg levels. 

### 3.2. THg Concentrations with Respect to Selenium and Other Elements, Age, and Body Condition

Se, an essential micronutrient, has been implicated in the reduction of THg toxicity with high molar ratios of Se/THg, considered protective [19,20,57,58]. Se concentrations and growth, as determined by fork length, weight, and linear growth, were well-correlated in Arctic char and even more obviously in lake trout (Table 4). In cisco, Se, and not age, was the greatest explanatory variable for THg (F(1, 97) = 15.46, *p* < 0.000), accounting for 14% of the variation (Table 3). Indeed, the relationship between Se and THg concentration was positive and significant for cisco as well as Arctic char (F(1, 196) = 5.87, *p* = 0.016), with average Se concentrations in the latter fish being 1.4-fold higher than levels in the other three salmonids (Table 2), possibly reflecting the absolute dependence of these Arctic char on a summer marine-based diet [54]. Consistent with this suggestion, local IK teaches that Arctic char can be caught in nets overwinter and not on baited hand lines under ice like some of the other salmonids, presumably due to seasonal fasting. Se bioaccumulates at lower levels in fish that have a freshwater-based diet, even if they feed in the sea in the summer, and thus the fast-growing lake trout may not be able to completely mitigate Hg exposure, notwithstanding their high Se/THg ratio. Nonetheless, the levels of Se in all the salmonids under study appear to underscore their anadromous life history rather than a strictly freshwater habitat, and with molar Se/Hg ratios at 8–9 for the *Coregonus* species and 14 and 25 for Arctic char and lake trout respectively, there could be some attenuation of toxicity in all these fish with ratios consistent with those reported for anadromous species in the Russian Arctic [20].

Differences in THg accumulation were also sought in secondary explanation variables. After age, condition factor (K) accounted for the greatest variance of THg in lake trout (F(1, 135) = 22.80, *p* < 0.000; Table 3). This significant relationship indicated that fish with lower K values, reflecting reduced body condition, had higher THg levels. This also appeared to be true, but to a lesser extent, in Arctic char (F(1, 196) = 20.99, *p* = 0.000), but not so clearly in the *Coregonus* species (Appendix A). As fish age, their body condition can decline, which can result in both advanced age and low K being linked to higher THg accumulation. However, specific comparisons with same year-class fish with different THg levels were illuminating. For example, of two 30-year-old lake trout from the same brackish-water site (Backhouse Point), one with poor calculated condition (K = 0.9) had high Hg levels (THg = 0.54 µg·g^−^^1^ ww), compared to a lake trout of the same age in good condition (K = 1.3) with low Hg levels (THg = 0.09 µg·g^−^^1^ ww). Another example is afforded by two 13/14-year-old Arctic char from the same freshwater site on Koka Lake, one with poor condition (K = 0.9) and a high THg level (0.35 µg·g^−^^1^ ww) vs. the other with good condition (K = 1.1) and a low THg level (0.02 µg·g^−^^1^ ww). Strikingly, both these comparisons align with the toxicity threshold of ~0.2 µg·g^−^^1^ ww THg, which is defined based on sub-lethal endpoints of growth, development, and behavior in fresh- and salt-water fish, with a lower threshold for reproductive effects [59,60,61,62]. For the Arctic char pairs, the fish with the poorer condition had a shorter overall fork length (~60%), but for the lake trout pairs, this relationship was reversed, with the fish with the higher K value having a reduced length (~80%), again reinforcing our contention that THg bioaccumulation is not strictly tied to fork length. Taken together, the low K values in these individuals suggest that high levels of THg could impact fitness and be associated with reduced foraging success. 

The possible role of seasonal diet in THg bioaccumulation led to the measurement of carbon and nitrogen isotopes to investigate the impact of proxy values for trophic position (δ^15^N‰) and forage location (δ^13^C‰). Results for these isotopes in Arctic char and lake trout were very similar to those reported for anadromous populations in the west Kitikmeot region of the Arctic [29]. These two species did not differ significantly from each other in mean δ^15^N values (Table 2), but notably, Se concentrations explained 46% of δ^15^N levels in lake trout and much less in Arctic char (Table 4). There was a significant positive relationship with trophic position in relation to changes in age, fork length, and mass in each of the salmonids, in addition to linear growth in Arctic char, lake trout, and cisco (Table 5). A correlation with growth is in accordance with the typical δ^15^N enrichment increases as fish grow and eat larger prey, with an increase of ~3.4‰ in δ^15^N corresponding to a shift of one trophic level [51,63,64,65]. Indeed, the *Coregonus* species showed lower δ^15^N enrichment, comparable to the 3.4‰ estimate representing a full trophic level [62]. In contrast, lake trout are piscivores for the majority of their lifetime and consume other salmonids including trout [28], and this is likely why lake trout showed a lesser dependence on age with trophic position than for the other fish (Table 5, Appendix A). In particular, although Arctic char had a similarly high δ^15^N enrichment as lake trout, the food web position of char appeared to change somewhat with age, likely reflecting a shift from an invertebrate-dominated diet of amphipods, copepods, cladocerans, and insects, to piscivory ([66,67,68], Table 2 and Table 5, Appendix A). Thus, in Arctic char, age likely explains the association with trophic position and THg (F(1, 101) = 9.03, *p* = 0.003), but since lake trout and Arctic char had similar trophic positions but substantially different THg levels, this emphasizes our argument that trophic position by itself cannot explain THg accumulation.

We considered that another explanatory variable for the low THg level in Arctic char could be assessed using δ^13^C (‰), or carbon sourcing (foraging habits). Fish that feed in the benthic zone tend to have higher δ^13^C than those that forage in the offshore, pelagic zones [69]. To understand food web relationships, δ^13^C is typically normalized to lipid content, and as previously reported, lipid assays of a subset of the samples (*n* = 101) showed that Arctic char was less fatty than the other three groups of salmonids that were all similar [18], and therefore, δ^13^C values can be directly compared in lake trout, lake whitefish, and cisco. Efforts to increase the overall sample size by using the C/N ratio as a proxy for lipid assays, as used by other researchers [26,70], could not be followed since there was no correlation in our samples between these two measures (not shown; consistent with [71]). Similar carbon signatures for Arctic char and cisco compared to lake trout and lake whitefish (Table 2) are therefore likely associated with benthic feeding [28,67,72]. Resource partitioning, or the relationship between relative trophic position (δ^15^N) and carbon sourcing (δ^13^C), was modeled for each of the fish groups (Appendix A). Changes in δ^15^N (‰) were significantly and positively related to changes in carbon sourcing for lake trout (F(1, 78) = 8.22, *p* = 0.005), not significantly related in Arctic char and whitefish, and significantly negatively related for cisco (F(1, 64) = 7.37, *p* = 0.008). It has been reported that at least for Arctic char, δ^13^C levels vary depending on the year, thus indicating broad prey choices [32]. Overall, our results suggest that resource partitioning is not sufficient to distinguish THg levels, and further indicate that both dietary pathways and trophic position can impact contaminant concentrations but do not seem to act independently.

When the salmonids that had been assessed for most parameters were examined together, without regard for species, correlation analysis showed that multiple factors correlated with THg, including age, length, weight, and K value, and many other elements (Figure 2). Spearman rank correlations were positive (*p* < 0.0001, unless stated otherwise) between THg and Rb (ρ = 0.530), Tl (ρ = 0.361), Ti (ρ = 0.154, *p* = 0.006), Mn (ρ = 0.235), Zn (ρ = 0.215, *p* = 0.00011), Ca (ρ = 0.151, *p* = 0.007), Mg (ρ = 0.288), P (ρ = 0.283), K (ρ = 0.325), Na (ρ = 0.372), and S (ρ = 0.449), and negatively correlated with Ni (ρ = −0.237), Cr (ρ = −0.288), and Cu (ρ = −0.262), but not with As, Sr, Co, Fe, or Se. The association between these various factors in the sampled salmonids was further investigated with exploratory PCA. As might be expected based on the correlation analysis, there were associations between THg and other elements: the non-essential Tl and Rb, in addition to the essential nutrients K, S, Mg, and P, plotted in the same quadrant when the fish were analyzed together (Appendix A). We are unaware that these relationships have been previously explored, with our results suggesting that factors including age that impact the uptake or retention of these various listed elements may be similar to those for THg accumulation. This further implies some independence of THg from species-specific foraging in salmonids. 

After samples were distinguished according to species in a factor loading plot that accounted for 43% of the variability shown in the PCA, distinct differences were apparent between Arctic char and lake trout, as well as the close proximity of cisco and lake whitefish (Figure 3). This may not be surprising since like THg, some other elements can bioaccumulate, with levels influenced by their properties, concentrations of other elements, the trophic level of the fish, as well as the habitat [71,72,73,74,75]. Thus, the PCA again suggests that factors associated with the uptake and retention of THg are similar for many other nutritional or essential elements. As noted, one such factor could be habitat, and as seen on different Alaskan islands, fishing site location can influence the levels of several elements [76]. Similarly, taking an example from our analysis, there was a clear overlap in the PCA between lake whitefish and cisco fished from the most northern fishing sites (Appendix A). In addition, the broad distribution of Arctic char from the southern island fishing sites suggests that characteristics of individual estuaries could have an influence on trace element accumulation since these fish do not feed while overwintering and would presumably spend the summers in the same sea waters.

### 3.3. THg Levels in East Kitikmeot Salmonids and Commercial and Subsistence Advisories

Although community members were interested in a comparison of THg levels in fish in different waters, this was complicated since fishing for some of the salmonids was not successful at all sites, particularly so for lake whitefish, which are at their northern-most limits [28]. In addition, Arctic salmonids may migrate away from the outlets of their overwintering waterways [77,78]. Previously, we had investigated if arsenic, polychlorinated biphenyls, and mercury contamination showed any site-specificity by dividing fishing sites on or near *Qikiqtaq* into groupings based on their geographic location (Figure 1, [18]). As noted, Arctic char had low THg levels independent of geographic grouping, ranging from 0.05 to 0.13 µg·g^−^^1^ ww, similar to *Coregonus* species (0.02–0.13 µg·g^−^^1^ ww), and in contrast to the consistently high THg values in lake trout (0.14–1.36 µg·g^−^^1^ ww, Figure 4, Appendix A). For popular island fishing sites, close to *Uqsuqtuuk*, lake trout from these locations (group 2, Figure 1) showed a mean THg concentration of 0.39 µg·g^−^^1^, within the range of the average levels for trout caught at sites on the mainland at 0.43 and 0.33 µg·g^−^^1^ ww, respectively (group 2 vs. groups 3–4, Appendix A). It is important to note, however, that island-sampled lake trout had a considerably younger mean age than the overall sampled population (14.6 vs. 27.8 years, respectively), likely reflecting increased fishing pressure due to their accessibility [79], but also increasing the unease about THg levels in lake trout at fishing sites close to the community. Although cisco from *Qikiqtaq* sites had significantly (*p* < 0.01) higher mean THg levels than those obtained from mainland sites (groups 1–2 vs. 3–4) at 0.16 and 0.08 µg·g^−^^1^ ww respectively, such concentrations are still low (Appendix A). While we are unaware of the status of permafrost degradation in this region, accumulated Hg and MeHg are released following permafrost thaw [80]. Therefore, there could be variation in THg accumulation depending on the aquatic ecosystem and landscape, as noted in other geographies [30,31,53]. However, our assignment of individual fishing sites into groups, required to maintain sufficient sample numbers, may have masked an effect at a particular site. 

As noted previously, although in lesser detail [18], of the 537 Arctic fish sampled, 31 fish exceeded Canadian mercury guidelines for commercial sale (0.5 µg·g^−^^1^ wet weight [81]), and 30 of these were lake trout, representing ~19% of the samples, ranging from 11 to 62 years of age. These data suggest that lake trout from this region could not responsibly be commercially sold. In contrast, since no Arctic char or cisco exceeded the commercial sale guideline, there should be no barrier to the community’s investigation of fisheries for Arctic char or lake whitefish ([82] and not shown). Of some concern, however, is that a greater number of fish exceeded advisories for subsistence consumption, designed to protect vulnerable people who rely on local and regional food fish. Such advisories have been issued for many countries, most states in the USA, as well as Canadian provinces and some territories (e.g., Alaska advisory for women and children at 0.15 mg·kg^−^^1^, sport fish in California at 0.2 mg·kg^−^^1^, the USA New England region at 0.2–0.3 mg·kg^−^^1^, and Alberta consumption limit advice for fish levels between 0.2 and 0.5 mg·kg^−^^1^ [83,84,85,86]). Of the 126 exceedances of advisory subsistence levels set elsewhere corresponding to ~0.2 µg·g^−^^1^ ww (Figure 4, see arrows), 99 were lake trout, representing 70% of all trout sampled. This advisory guideline was surpassed even by the youngest sampled age (8 years). The relationship between THg and otolith-assigned year class for all 537 individual samples is shown (Figure 4), and for lake trout, they are distinguished as being caught at mainland sites (individuals as black dots) and Qikiqtaq sites (island-caught fish as red dots). As noted, the island fishing sites are favored particularly by community members who do not own boats or depend on the community Hunters and Trappers Association’s “food bank”, as well as fish caught during the regular trout ice-fishing derbies. Regrettably, as indicated, lake trout from the island generally do not appear to have lower THg levels than those from mainland sites (Figure 4, Appendix A). Concerningly, and as noted, the island lake trout are about half the average age of the mainland samples and therefore might have been expected to have lower THg levels, considering that age explains almost ⅓ of the THg variation in this species (Table 3). Additionally, while it has been reported that exposure to THg can be mitigated when fish are cooked [87], Inuit from this community do consume fish raw, particularly when “out on the land”. We are not aware of any Government of Nunavut fish consumption advisories, and ethics permits to conduct human consumption studies for this purpose should be spearheaded by the community and health authorities. However, considering the popularity of lake trout among many *Uqsuqtuuk* residents, we suggest that the territorial government develop specific consumption guidelines in close collaboration with communities and their health units for this salmonid in order to prevent exposure to high levels of MeHg, a neurotoxin, in pregnant women and in children. As noted, all the other salmonids tested had THg levels that were generally below subsistence level advisories as well as Canadian commercial guidelines, and thus with respect to this contaminant, the consumption of Arctic char, lake whitefish, and cisco should be encouraged. 

## 4. Conclusions

As indicated by a wealth of literature, in addition to this study that investigated multiple influences on the levels of THg, the bioaccumulation of this element depends on many biotic and abiotic factors, including age, growth rates, condition, trophic level, and concentrations of selenium and other elements (Appendix A), with the association of some of these factors also influenced by species and to a lesser extent by geography. Our analysis can be distinguished from previous work in that anadromous Arctic char, lake trout, cisco, and lake whitefish from this understudied Arctic region were sampled on many of the same migratory runs, showing that even under similar conditions, and with similar physiological changes, contributary factors vary the bioaccumulation of THg in concert with certain other elements. THg levels are clearly influenced by phylogenetics, as shown by the PCA that highlighted the segregation of different species into different quadrants. Analysis of the genetic and microbiome differences [18,54,55] between these salmonids may be a future promising avenue to reconcile some of the disparities in our understanding of tissue concentrations of THg in different salmonids. In addition, it is our hope that the *Uqsuqtuuk* community will continue to access Arctic char, lake whitefish, and cisco as healthy food choices and as a resource for economic benefits. Nevertheless, our data underscore a concern about the advisability of sourcing lake trout for community prenatal cooking classes, the “mums and tots” group, school food programs, as well as celebratory events and community feasts. Due to the special vulnerability of Arctic regions to atmospheric Hg emissions and thawing permafrost, we further recommend that monitoring of these food fish should continue on a periodic basis, both for the long-term health of subsistence consumers, and for the well-being and safety of any fishery operations.

## Figures and Tables

**Figure 2 foods-10-02621-f002:**
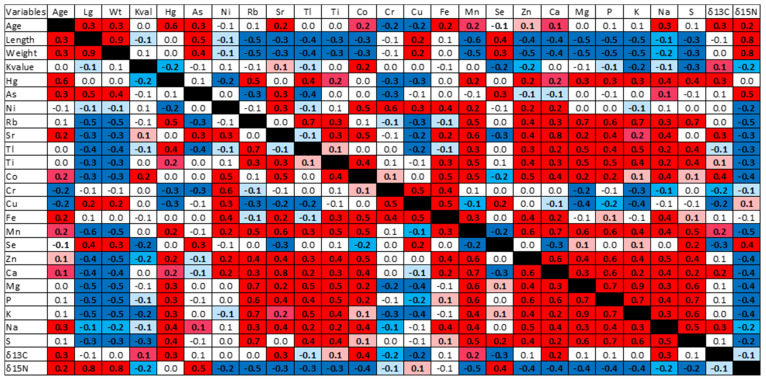
Spearman’s rank correlations were calculated between elements at sufficient concentrations to be detectable in >70% of salmonid samples (including THg), as well as age, as determined by otolith analysis, fork length (Lg), weight (Wt), condition (Kval), elements (as shown by their standard abbreviations), as well as levels of isotopes for carbon (δ^13^C) and nitrogen (δ^15^N). The correlation matrix (DF = 319) shows rho (ρ) values in cells. Statistically significant correlations are colored, with colors indicating *p*-values < 0.001 (red for positive correlations, dark blue for negative correlations), *p* between 0.001 and 0.01 (pink for positive, aqua for negative), and *p* between 0.01 and 0.05 (pale pink for positive, pale blue for negative).

**Figure 3 foods-10-02621-f003:**
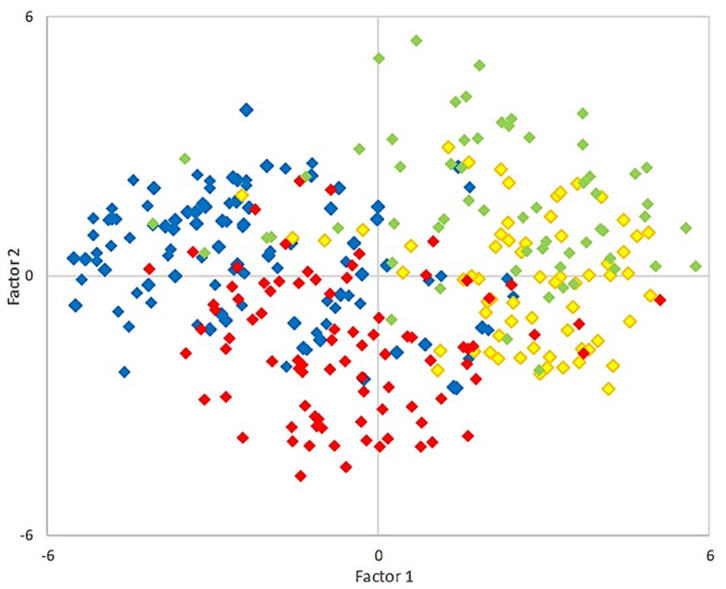
Principal component analysis using the data and parameters in Figure 2, showing the first two factors (factor 1 on the x-axis and factor 2 on the y-axis at 29.8% and 13.5% respectively, for a sum of 43% of the variance) with the different salmonids’ samples plotted so that the segregation of the different species with respect to these factors can be more easily visualized. Arctic char samples (blue) cluster in the top left quadrant, lake whitefish and cisco on the right side (light green and yellow, respectively), and most of the lake trout (red) plot in the lower portion. Appendix A shows the same plot but with the different fish showing different symbols representing the sampling location according to each geographical grouping, as shown in Figure 1.

**Figure 4 foods-10-02621-f004:**
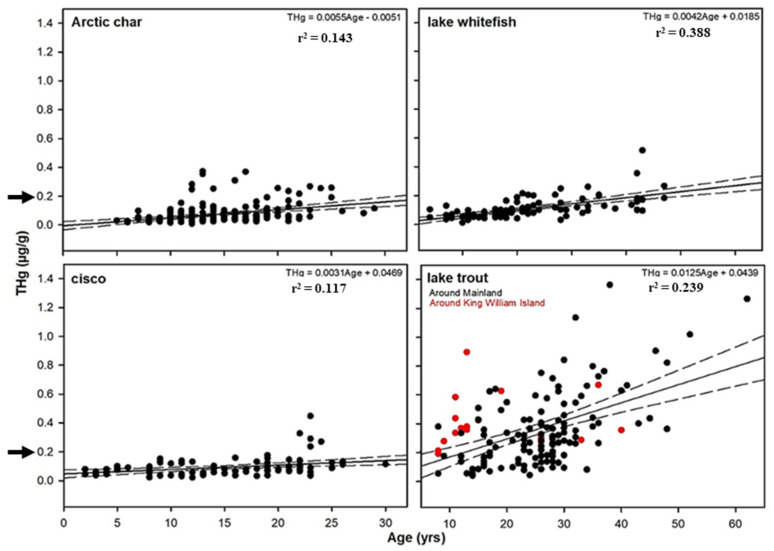
Relationship between total mercury (THg; µg·g^−^^1^) and age (years, determined by otolith analysis; note the different scale for ages of Arctic char and cisco vs. lake whitefish and lake trout) with regression lines and with dots indicating individual fish samples, albeit with many overlapping dots. The recommended level for THg concentrations in fish for subsistence fisheries from several jurisdictions (see Results and Discussion Section) is approximately 0.2 µg·g^−^^1^ (shown as black arrows on the labeled axes). For lake trout, THg levels in fish caught on King William Island are shown as red dots, with all of these falling at or above the 0.2 µg·g^−^^1^ level. Further analysis showed that these were not statistically lower than comparable fish obtained from the mainland, which are shown as black dots (see Results and Discussion Section). The regression line represents lake trout from the mainland with the corresponding r^2^ noted, with the island (red dots) regression line not shown with r^2^ = 0.324, and for all total lake trout assayed as indicated in Table 3.

**Table 1 foods-10-02621-t001:** Fishing sites sampled in the east Kitikmeot region of Nunavut, Canada, showing names of fishing sites (English and Inuktituk), global positioning system (GPS) coordinates, water type, and fishing gear used.

Site (English Name)	Site (Inuktitut Name)	GPS Location	Water Type and Fishing Gear
Port Parry	Tununik	N 69°33′28.764″ W 97°26′13.884″	fresh/nets
Koka Lake	Koka	N 68°32′5.1″ W 96°12′45.899″	fresh/nets
Backhouse Point	Kautakshat	N 67°27′27.2″ W 95°21′38.6″	brackish/nets
Legendary River	Kuuktaq	N 67°31′17.8″ W 96°26′21.8″	brackish/nets
Merilik Lake	Malirualik	N 68°34′25.98″ W 97°19′36.72″	fresh/nets
Murchison River	Kuuk	N 68°25′35.5″ W 93°19′11.6″	fresh/nets
Back River	Amurat	N 66°57′30.70″ W 95°18′5.20″	fresh/nets
KWI Weir	Iqalugmiat	N 68°55′58.08″ W 96° 13′10.2″	fresh/spears/nets
West of Chantry 3	Panigtogaruk	N 67°48′41.7096″ W 97°2′37.457″	fresh/nets
West of Chantry 4	Tahuaruak	N 67°52′10.4736″ W 96°43′7.3128″	fresh/nets

**Table 2 foods-10-02621-t002:** Sample size, morphometric data, and age, as well as mercury, selenium, and stable isotope ratios obtained from muscle samples derived from four salmonid species caught in the Kitikmeot region of Nunavut, Canada, with the number of fish sampled shown as *n* (except for the stable isotope analysis, which is adjacently noted), and with units indicated in parentheses and calculations shown as means and 95% confidence intervals (CI).

Species				
Variable	Arctic Char	Lake Trout	Lake Whitefish	Cisco spp.
Sample size (*n*)	197	136	100	98
Fork length (mm)	624 ± 18	638 ± 20	398 ± 10	334 ± 16
Body weight (g)	3086 ± 247	3308 ± 186	935 ± 79	504 ± 66
Condition factor (K)	1.155 ± 0.037	1.163 ± 0.028	1.409 ± 0.043	1.162 ± 0.057
Age:				
Range	5–29	8–62	4–47	2–30
Mean ± 95% CI	14.2 ± 0.6	25.4 ± 1.6	21.3 ± 2.2	14.8 ± 1.4
Linear growth (mm∙year^−1^)	47.1 ± 1.9	27.8 ± 1.4	23.8 ± 2.8	28.5 ± 2.9
Mercury content (µg∙g^−1^ ww)	0.073 ± 0.009	0.350 ± 0.042	0.092 ± 0.012	0.092 ± 0.013
Selenium content (µg∙g^−1^ ww)	0.44 ± 0.02	0.32 ± 0.02	0.34 ± 0.06	0.31 ± 0.02
δ^13^C content (‰)	−25.1 ± 0.4	−23.9 ± 0.7	−21.3 ± 0.6	−25.3 ± 0.5
δ^15^N content (‰)	14.0 ± 0.3	14.3 ± 0.3	11.3 ± 0.3	11.8 ± 0.2
Assayed for δ^15^N & δ^15^N (*n*=)	102	79	55	65

**Table 3 foods-10-02621-t003:** Coefficient of determination (r^2^) showing the proportion of variation in total muscle mercury (THg) concentrations in relation to age, fork length, body weight, condition, and linear growth, as well as soft tissue concentrations of selenium and stable carbon and nitrogen isotope ratios with statistical significance of the regressions, noting that negative relationships are indicated by minus signs.

Species	*n*	Age (years)	Fork Length (mm)	Round Weight (g)	Condition Factor (K)	Linear Growth (mm∙year^−1^)	Selenium (µg∙g^−1^ ww)	δ^13^C (‰)	δ^15^N (‰)
Arctic char	197	0.143 **	ns	ns	−0.097 **	−0.042 **	ns	0.120 **	−0.083 **
Lake trout	136	0.300 **	0.088 **	ns	−0.145 **	ns	ns	0.098 **	ns
Lake whitefish	100	0.388 **	0.113 **	0.074 **	ns	ns	ns	ns	ns
Cisco spp.	98	0.117 **	ns	ns	ns	ns	0.139 **	ns	ns

** *p* ≤ 0.05, ns = not significant.

**Table 4 foods-10-02621-t004:** Coefficient of determination (r^2^) showing the proportion of variation in soft tissue selenium concentrations in relation to age, fork length, body weight, condition, and linear growth, as well as stable carbon and nitrogen isotopes with statistical significance of the regressions, noting that negative relationships are indicated by a minus signs.

Species	*n*	Age (years)	Fork Length (mm)	Round Weight (g)	Condition Factor (K)	Linear Growth (mm∙year^−1^)	δ^13^C (‰)	δ^15^N (‰)
Arctic char	197	0.058 **	0.149 **	0.055 **	−0.063 **	0.111 **	−0.162 **	0.053 *
Lake trout	136	ns	0.195 **	0.289 **	0.099 **	0.127 **	ns	0.458 **
Lake whitefish	100	ns	ns	ns	ns	ns	ns	ns
Cisco spp.	98	−0.054 *	ns	−0.055 *	ns	ns	ns	ns

** *p* ≤ 0.01; * *p* ≤ 0.05; ns = not significant.

**Table 5 foods-10-02621-t005:** Coefficient of determination (r^2^) showing the proportion of variation in carbon and nitrogen isotopic concentrations in muscle tissue in relation to age, fork length, body weight, condition, and linear growth, and showing statistical significance of the regressions (with negative relationships indicated by minus signs), noting that several coefficients provided were not significant but approached significance—these and probability levels are referenced by superscripts and provided as footnotes.

**Isotopic Carbon–δ^13^C**							
**Species**	** *n* **	**Age** **(years)**	**Fork Length** **(mm)**	**Round Weight** **(g)**	**Condition Factor** **(K)**	**Linear Growth** **(mm∙year^−1^)**	**δ^15^N** **(‰)**
Arctic char	102	ns	ns	−0.032 ^a^	ns	−0.073 **	ns
Lake trout	79	0.122 **	0.086 **	ns	−0.082 *	ns	0.097 ^b^
Lake whitefish	55	0.206 **	0.069 ^c^	0.067 ^d^	ns	ns	ns
Cisco spp.	65	ns	ns	ns	0.109 **	ns	−0.105 **
**Isotopic Nitrogen–δ^15^N**							
**Species**	** *n* **	**Age** **(years)**	**Fork Length** **(mm)**	**Round Weight** **(g)**	**Condition Factor** **(K)**	**Linear Growth** **(mm∙year^−1^)**	
Arctic char	102	0.193 **	0.567 **	0.437 **	ns	0.406 **	
Lake trout	79	0.109 **	0.456 **	0.502 **	ns	0.377 **	
Lake whitefish	55	0.191 **	0.111 *	0.084 **	ns	ns	
Cisco spp.	65	0.253 **	0.617 **	0.414 **	ns	0.088 *	

^a^ *p* = 0.070; ^b^ *p* = 0.053; ^c^ *p* = 0.053; ^d^ *p* = 0.057; ** *p* ≤ 0.01; * *p* ≤ 0.05; ns = not significant.

## Data Availability

The data have been deposited in the Polar Data Catalogue (PDC) as open access to all polar researchers (PDC#312992; NA profile of IOS 19115:2003, uploaded 5 February 2020, doi.org/10.21963/12992).

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
