# Peer review of "Correlation of Mercury Occurrence with Age, Elemental Composition, and Life History in Sea-Run Food Fish from the Canadian Arctic Archipelago’s Lower Northwest Passage"

_foods, 2021, doi:10.3390/foods10112621_

Round 1

Reviewer 1 Report

     Koch et al. submitted a manuscript on characterizing concentrations of mercury (Hg) and multiple other elements in fish harvested from the Northwest passage. The authors have done extensive work and analysis to understand factors correlate with the various elements. This manuscript overall is interesting and adds to the literature on the topic. This reviewer only has experience with elemental analysis of foods and cannot speak to the other areas of the manuscript.

     It is noted that the authors have produced a large amount of data from these experiments and would like to thoroughly analyze the data to detect any potential interesting findings. However, a good portion of the manuscript comes across as a “fishing expedition” with only limited rationale for some of the analyses. For instance, the introduction goes into depth describing how potential thawing of the permafrost driven by higher regional temperatures can release Hg into the fish’s habitats, leading to higher Hg residues in the fish. This is an interesting hypothesis, but the current experiments seem in no way to test this research question. In addition, the authors briefly mention that it is unknown whether the areas where the fish were harvested are near areas with thawing permafrost. It is suggested that the authors pin down the most important findings in the manuscript for clear communication.

It was surprising that the differences in Hg species were glossed over in the introduction. Chemical differences in inorganic and organic species should be mentioned, especially how methylation of Hg will increase the ability for it to partition into lipids.

Comments that should be addressed appear below:

ABSTRACT

  • Title is unclear. Suggest instead “…Northwest passage: Correlation of mercury occurrence with fish age and elemental composition.”  
  • Line 23. Please put -1 in superscript (and throughout abstract)
  • Line 26. The conclusion proposes monitoring of the fish for Hg due to the potential for vulnerable populations exposed to this metal. However, much of the manuscript does not speak to how the levels of Hg exceed regulatory limits in foods. Surprisingly, this information appears only in the supplementary information. This should be better emphasized if the authors are using this as the main conclusion.

INTRODUCTION

  • The introduction could be shortened a bit and repositioned to be more focused on the topic at hand. Also to note is that there are over 80 references for this paper. This is quite high for such a paper and there should be a focus on only the most important references. (20+ could easily be eliminated.)
  • There are various regulatory limits on Hg in fish for different countries. It would be worth mentioning these so the reader can get an idea in the introduction on the governmental efforts to limit Hg residues in the foods supply.
  • Line 44. Please expand on speciation of inorganic versus organic Hg.
  • Line 52. What is meant buy the protection of Hg toxicity by Se? Is this for the fish or for people who consume fish contaminated with Hg? Please explain.
  • Line 34. Please put 2+ for Hg (missing +).

MATERIALS AND METHODS

  • Line 115. This section on lab analysis should be divided to better separate elemental analysis and other aspects of determining the physical characteristics of the fish.
  • The authors need a section on chemical reagents used or they need to better describe the ICP-OES and -MS analysis used. What type of acids were used? Was HCl added to stabilize the Hg species (as they are volatile). What internal standards were used? What measures were taken to prevent elemental contamination? It appears that glassware was used, which can introduce elemental contamination. Please clarify.
  • Line 172. What does biological data refer to? How was data normality assessed? Please specify. Line 184. Were there differences in the manner elements with detectable versus quantifiable levels of elements were analyzed? How was data treated if it was detectable but below the LOQ? The authors should add supplementary information on the analyte LOD/LOQ and how they were determined
  • Line 197 and throughout section. Please place -1 in superscript.

RESULTS/DISCUSSION

  • Line 213. What does the infinity sign refer to?
  • Table 3 and throughout. For age, it is specified that the error is showing the 95% CI. However, it is not specified for other parameters.
  • Figure 2. It is difficult to understand how these correlations are presented. The caption says that these correlations are with THg, but looking at the figure it appears that it is showing the correlation of the elements/parameters along the border of the table. For instance, the intersection of As and age is 0.3. How does this relate back to THg?
  • Figure 3. The caption does not mention the input data for the PCA plot. Was this for the elemental data?
  • Table 3 and Figure 4. How do these differ other than showing a visual versus tabular version of the data?
  • For the conclusion, it is assumed that these fish are consumed in large amounts, especially at certain events. However, the authors did not cite information on any intake estimates of these fish or perform a risk assessment. Are there any consumption data available? Also, the authors need to focus more on the actual Hg content of the fish and how it related to regulatory Hg limits in foods.

Author Response

Please see attachment (response to both reviewers, with thanks)

Reviewer 2 Report

It seems to me an interesting and well-developed work, although it is limited to a very specific geographical area, which is
why I have missed that in the discussion of results the authors have not developed specific consumption guidelines for this salmonid in
order to prevent exposure to high levels of MeHg, especially in vulnerable populations such as pregnant women and young children. I think it would be interesting to specify in the research work, the consumption patterns to be followed by these population groups.

Author Response

Please see attachment (reports from both reviewers since there was overlap, and with thanks)

Round 2

Reviewer 1 Report

The authors have made substantial revisions to the manuscript and have thoroughly addressed reviewer's concerns. This will make a interesting contribution to the research field.

A few items to clarify:

-line 14 in Abstract. Is it assumed Hg emissions are from industrial sources?

-line 17. Add n= before number.

-line 175. It seems that "Analysis of" should be added before "Stable"?

Author Response

Referee #1   

The authors have made substantial revisions to the manuscript and have thoroughly addressed reviewer's concerns. This will make a interesting contribution to the research field.

A few items to clarify:

-line 14 in Abstract. Is it assumed Hg emissions are from industrial sources?

This is correct but the limit on the abstract is 200 words, which has been reached so further elaboration in this section is not possible.

-line 17. Add n= before number.

Done

-line 175. It seems that "Analysis of" should be added before "Stable"?

Good suggestion- thank you

Thank you again to this reviewer.